# Application of Feature Extraction Methods for Chemical Risk Classification in the Pharmaceutical Industry

**DOI:** 10.3390/s21175753

**Published:** 2021-08-26

**Authors:** Mariusz Topolski

**Affiliations:** Department of Systems and Computer Networks, Faculty of Electronics, Wrocław University of Science and Technology, Wybrzeże Wyspiańskiego 27, 50-370 Wrocław, Poland; mariusz.topolski@pwr.edu.pl

**Keywords:** industry 4.0, cybersecurity, communication, principal component analysis, imbalanced

## Abstract

The features that are used in the classification process are acquired from sensor data on the production site (associated with toxic, physicochemical properties) and also a dataset associated with cybersecurity that may affect the above-mentioned risk. These are large datasets, so it is important to reduce them. The author’s motivation was to develop a method of assessing the dimensionality of features based on correlation measures and the discriminant power of features allowing for a more accurate reduction of their dimensions compared to the classical Kaiser criterion and assessment of scree plot. The method proved to be promising. The results obtained in the experiments demonstrate that the quality of classification after extraction is better than using classical criteria for estimating the number of components and features. Experiments were carried out for various extraction methods, demonstrating that the rotation of factors according to centroids of a class in this classification task gives the best risk assessment of chemical threats. The classification quality increased by about 7% compared to a model where feature extraction was not used and resulted in an improvement of 4% compared to the classical PCA method with the Kaiser criterion, with an evaluation of the scree plot. Furthermore, it has been shown that there is a certain subspace of cybersecurity features, which complemented with the features of the concentration of volatile substances, affects the risk assessment of chemical hazards. The identified cybersecurity factors are the number of packets lost, incorrect Logins, incorrect sensor responses, increased email spam, and excessive traffic in the computer network. To visualize the speed of classification in real-time, simulations were carried out for various systems used in Industry 4.0.

## 1. Introduction

The modern economy is based on knowledge.

Various technologies for data processing and exchange, mutual automation and manufacturing techniques are used in many different areas of a company’s activity. The rapid pace of the technological progress contributed to the creation of the Industry 4.0 concept. This is the so-called fourth industrial revolution that increases the productivity of the modern economy. Such a concept is associated with an intelligent factory in which various systems control physical processes and make decentralized decisions. By using computer networks, these systems communicate with each other and with people [1]. Modern industry is developing dynamically, which can be seen through technological progress. The competitiveness of enterprises is closely related to adapting to the development process of the Industry 4.0 concept. One of the goals of the modern economy is to reduce production costs and pollution generated in production processes. Certain side effects of production pose a high risk of biological and chemical hazards. Therefore, there is a need to implement technologies supporting the management of harmful substances’ emission to the environment. The release of chemical substances into the air, in addition to the negative impact on the environment, poses a high risk of loss of health and life for employees of production lines in the pharmaceutical industry.

In the era of computerization progress, chemical risk management must be coordinated through the efficient introduction of new procedures, along with the changing [2,3,4] manufacturing technologies. Broadly understood, information flow management is a challenge for modern enterprises, especially those integrating with the concept of Industry 4.0. Risk management is the identification, assessment and measurement of risk. Finally, risk management measures are taken, especially in situations where it cannot be eliminated [5,6].

The modern economy processes large volumes of data. Various feature extraction methods are used to reduce the size of the data.

One of the key challenges in feature extraction is determining the number of features with strongest discrimination ability from the domain. Currently used approaches of estimating the number of components and features can be based, for example, on the Kaiser [7] criterion or scree plot [8]. Those methods are mainly based on eigenvalues. Their main drawback is often over- or underestimation of the number of components needed. This behavior can translate to worsening of the classification quality. Research on estimating the number of features is based primarily on the Kaiser Criterion and is based on evaluation of eigenvalues for which principal component analysis would allow for the highest percentage of explained variance [9,10,11,12]. In the literature, a curious model of empirical determination of the number of features based on the Kaiser criterion is presented in the work of Breaken and Aseen [13]. These authors, however, obtain the same feature space as with the classical Kaiser criterion, where only components having an eigenvalue >1 are taken [7,8].

The motivation behind this work is to propose an alternative criterion to the Kaiser and scree plot which, by allowing more precise feature extraction, would lead to an increase in the quality of classification. The purpose of the work is to develop and analyze a method for estimating the number of features and components for principal component analysis. The second objective is to apply the developed method to the chemical risk analysis in the pharmaceutical industry. The developed method allows for the extraction of features based on given data by determining the beta value for which the number of components and features most strongly discriminate against a given feature subspace. The second objective concerns the application of the developed method in the task of chemical risk classification in the pharmaceutical industry. For this goal, a dataset containing 40,000 examples or 63 features was used. The essence of the problem under consideration was the task of classifying risk of chemical hazards. This risk can be caused by two factors. The first is the measurement results from sensors of various concentrations of volatile substances produced during the production of pharmaceuticals. The second factor concerns the various anomalies present in the computer network.

The second section describes the chemical risk in Industry 4.0. The third section provides an overview of the literature from similar works. In the next section, the author’s proposed criterion is described. In the experimental part, the focus was on assessing the quality of classification by comparing the developed method with others known from the literature, the Kaiser criterion and scree plot. For this purpose, a chemical risk dataset was used, taking into account cybersecurity features and concentrations of volatile substances. Various classifiers and extraction methods were compared. The research part was divided into four experiments. In the last part, the obtained results were discussed.

## 2. Chemical Risk in the Pharmaceutical Industry

The pharmaceutical industry is a specific production environment. This is because various chemicals used in the production of drugs are produced there. They have different chemical and physical properties, and different levels of toxicity. The diversity and combinations of these substances is so great that it is impossible to list them all. Quite a big problem are fillers, i.e., inert agents aimed at improving the appearance, durability or supporting the solubility of the finished product [14]. Fillers pose a serious risk of poisoning workers, and the properties of these agents can lead to ignition or explosion. Serious chemical hazards in the pharmaceutical industry are rare. The frequency and effects of chemical hazards have increased significantly in recent years, which is also due to the development of the chemical industry. By industrial hazard, we mean a condition that causes human injury, disease or death [14]. There are hazards in the production of drugs with the following substances and mixtures:With explosive properties;With oxidizing properties;Extremely flammable;Highly flammable;Very toxic;Harmful;Corrosive, irritating item;Sensitizing;Mutagenic;Toxic for reproduction;Dangerous for the environment [15].

Figure 1 shows the algorithm of chemical risk estimation [15].

The abbreviations mean: Ps—the exposure index which enables the assessment of the weighted average concentration for the entire work shift; this indicator may be, in the case of measurements using individual dosimetry, the weighted mean concentration for the work shift (Cw), and in the case of stationary measurements, the upper limit of the confidence interval for the true mean (GG) or the upper limit of the confidence interval for the weighted mean concentration for the entire work shift. GGw—these values are given in the protocol for measuring harmful factors in the work environment, directly to be used in the risk level estimation. Pch—exposure index for the assessment of instantaneous concentrations, Pp—exposure index for the assessment of ceiling concentrations. Many substances are used to manufacture drugs. This results in complex emissions of harmful gases into the air. In this case, the value of the Ps index is calculated, which is the value of the sum of the values obtained for individual component TLV values [15]:(1)$$P_{s}=\frac{P_{s1}+P_{s2}+...+P_{sr}}{NDS_{1}+NDS_{2}+...+NDS_{r}},$$
where: $P_{sr}$—exposure index for *r* substances, $NDS_{r}$—the value of the highest *r*—permissible concentration values of the substance.

## 3. Related Work

The modern manufacturing industry generates large volumes of data. For that reason more and more complex data processing methods are used, i.e., machine learning methods [16]. Among such solutions in the Industry 4.0 concept, the concept drift analysis method can be used to improve cybersecurity anomaly detection systems [17], or the use of active learning strategies in non-stationary data streams [18]. The classification task uses an ensemble approach. In the literature, among others, the subject of linear combination of classifiers is discussed in this area [19].

Along with technological progress of Industry 4.0, the volume of data and features describing a given problem increase. This causes an exponential increase in the necessary experimental data that depend on space dimension, which is important in the pattern recognition task. In many practical cases, recognized classes are unbalanced, which might be caused by rare cases, e.g., disease disorders. Feature engineering is a field that aims to reduce the dimensionality, which simplifies not only decision-making process, but also time complexity of calculations and quality of classification.

The phenomenon we are dealing with is called the curse of dimensionality and it is associated with an increase of the number of features in relation to the slow increase in the number of objects. In order to reduce the dimensionality of data, various methods of selection and extraction can be used. With the selection of methods, we are able to select those features that carry important information [20,21].

Filters are the simplest group of data selection methods. The most commonly used methods of this type is ANOVA [22], i.e., analysis of variance, Pearson’s correlation coefficient [23,24] or the chi-square test [22,25].

Another group of analyzed methods is called wrappers, which make the selection according to the analysis of particular classifier performance. Wrappers select different subgroups of features from an available set and then use some portion of the dataset to check the classification results against these features. This process is repeated multiple times for the identification of features that have the greatest impact on the results obtained by a particular base classifier. It is worth noting that for different classification algorithms, the optimal set of features may differ. Such a selection, although often more effective than filters, has a certain disadvantage in the form of high computational load [21,22]. A complete review of all combinations of features is usually impractical or impossible to perform. For this reason, wrappers use various heuristics to improve the search for optimal subsets, such as the ant algorithm [26], the genetic algorithm [27] and others [28].

Embedded methods are another example of data selection methods. They get their name from the fact that they are built into the classifier itself. This means that features are selected during the classifier training process. It can be said that embedded methods combine the advantages of filters and wrappers. Features are selected based on their interaction with the currently used classifier. Their computational cost is reduced by combining the selection with the training process [21]. The fact that the selection mechanism is coupled with the classifier training process also offers new possibilities, for example, a special adaptation to unbalanced data [25].

Feature extraction is a more complex issue and its assumption is to discover important connections in the original data, and transfer them into a new space, where they can be represented in a smaller volume. This means that, unlike selection, extraction leaves no features in their original form. Instead, it creates new ones representing similar knowledge in a more compressed way [21]. Feature extraction can usually reduce the dimensionality of data more than selection. For this reason, it is often used to analyze data with many dimensions, such as images [29].

The most popular feature extraction method is Principal Component Analysis, which represents an unsupervised approach. In this method, the input dataset is presented as a cloud of points in a coordinate system with a number of dimensions corresponding to the number of features. This coordinate system is later rotated to maximize the variance of successive coordinates. The converted coordinates are a new input vector for the classifier [30].

Among the numerous applications of the PCA method, the analysis of hyperspectral images [31,32] can be distinguished. There are also modifications to PCA and attempts to combine it with, for example, Independent Component Analysis [33,34]. One of the PCA modifications is Centroid Class Principal Component Analysis. The advantage of this approach is that the rotation of the components takes place according to centroids of each class, which makes it possible to better match the features and patterns in the classification task [35]. Another modification is Gradient Stochastic Principal Component Analysis, which optimizes the selection of features for a given principal component based on the stochastic gradient [36]. The Kernel Principal Component Analysis is an extension of PCA, which uses nonlinear kernel transform functions and provides an alternative to linear relationships [37].

Another solution used in the feature space dimension reduction problem is Linear Discriminant Analysis. Its goal is to find a linear combination of features that will best distinguish the classes. The applications of LDA are very wide, from image analysis where it most often achieves results similar to PCA [38], or audio signals [39].

During the analysis of the extraction task, one should also mention the role of neural networks, the so-called autoencoders [40,41]. Inside such a network, the input vector is modified and transformed to obtain a vector with fewer elements. Then, the following layers of the network try to recreate the original vector as accurately as possible. This forces the network to contain as much information as possible in the reduced vector.

The Principal Component Analysis method, as well as its modifications described earlier, are used to reduce the size of the data by selecting only those features that contain the most information. However, the mentioned method has some disadvantages, i.e., the difficulty in the estimation of the number of components and features. There is no solution for the assessment of a given component when it has similar eigenvalues in various components. This is especially important in the process determination of the number of components, where utilization of different criteria may either cause too many or too few components.

## 4. Methods

In the classic PCA approach, the number of components and features is determined arbitrarily by using, for example, the K—Kaiser or the SP—scree plot criterion. This approach, although widely used, carries errors related to the selection of an appropriate number of features and components. In this paper, I present a method that solves the above mentioned problem.

Let *M* be a number of selected components and *n* a number of all features. For the selected *M* components according to the PCA, we obtain the following eigenvectors:(2)$$\forall m\in M\;W_{m}=\left[w_{m,1},w_{m,2},...w_{m,n}\right].$$

Let $\Phi (W_{m})$ be the standard normal distribution function N(0,1). The original approach to the task of determining the number of features and components is to use the reliability criterion given by the formula:(3)$$\beta =\frac{k\bar{r}\dot{r}}{1+\left(k-1\right))\bar{r}\dot{r}},$$
where: *k*—number of scale features, $\bar{r}$—average correlation coefficient between all pairs of items, $\dot{r}$—coefficient of determination defining the power with which a given feature explains the entire construct of variables. The $\dot{r}$ coefficient is the correlation coefficient between each feature separately and the summed scale of all features. Ultimately, the fact that each feature belongs to each *m*-component is determined by the fulfillment of the relationship:(4)$$\Phi (w_{m,i})\geq \beta \;i=1,2,...,n.$$

In a situation where a single feature has been classified to more than one component, belonging to a given component is determined by the criterion $max[\left|w_{m,i}\right|]$. For components explaining a progressively smaller percentage of the total variance, the number of features belonging to these components decreases and when Equation (Equation 3) is not fulfilled, such components are removed.

Let us consider an example of a nonlinear main component separation situation. Equation (Equation 3) may be insufficient. For a good separation, some margin $\left[\beta_{-};\beta_{+}\right]$ must be set with a nonlinear kernel function. A good solution to this problem is to use the nonlinear SVM [42] model. In this form, the PCA method becomes supervised. In such a task, in the Ω data space, the vectors $\beta_{i}$ are defined and create a teaching sample *D* belonging to two classes:(5)$$D={\left\{\left(\beta_{i},c_{i}\right)|\beta_{i}\in \left[0;1\right],c_{i}=\left\{-1;1\right\}\right\}}_{i=1}^{N}.$$

The task is to determine a classifier that allows to divide the entire Ω space into two separate areas corresponding to the −1, 1 classes and to classify features to principal components as good as possible. For this purpose, as already mentioned, the nonlinear support vector machine approach [42] can be used. If $K\left(x,\beta \right)$ denotes a symmetric function of two kernel vectors such that x∈X, $\beta \in \left[0,1\right]$ and X∈R, then the transform can be specified as ϕ:X↦Θ, where:(6)$$K\left(x,\beta \right)=\phi \left(x\right)\cdot \phi \left(\beta \right).$$

The Θ space, to which the mapping is done, is called a transformed variable space.

The final algorithm is Algorithm 1.
**Algorithm 1:** Model for determining the number of features *n* and components *M*.1Load dataset x∈X and normalize it—*N*(0; 1) 2Initialize methods PCA [37], CCPCA [35], GPCA [36] and KPCA [43]; 3Calculate the beta β: 4**for** i∈{1,⋯,n} **do**
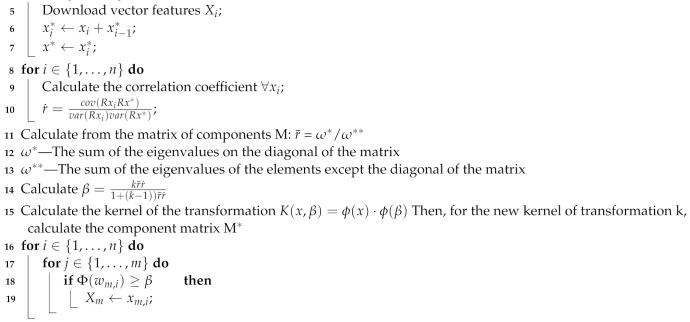


Figure 2 shows the principle of assigning features to one of the two components according to Equation (Equation 3) for a linear and nonlinear separation space.

The second important part of the article is to propose a data processing model by various pharmaceutical companies with the use of an integrated risk management system based on machine learning mechanisms. Its architecture is shown in Figure 3.

The basic assumption of the model is that each pharmaceutical company implements the Industry 4.0 strategy in the production of drugs and uses one or many of the different concepts of horizontal communication, i.e., Cloud Computing [44], IIoT [45], Connectivity [46], Big Data [47], Digital Twins [48], ID/Loc [46,49], Semantics [50,51], SOA [52]. Of course, we can assume other solutions, but the eight mentioned are key concepts of the CPS I4.0. The model proposes that the data exchange should take place using the CML (Chemical Markup Language) standard. Information about the values of the measured characteristics is stored in the main common repository. Then, the collected data are used by the Data Mining module, which also extracts features apart from classification. Each enterprise uses a common data repository and machine learning mechanisms. In this way, by sending data for classification in real time, the company receives a response within a certain time. This time is an important element of research. Because it determines the speed of reaction to the dynamically changing risk of chemical hazards.

## 5. Experimental Set-Up

The research was divided into four separate experiments to answer three key research questions:

RQ1: How does the proposed method of determination of the number of components and features work against the criteria known from the literature, taking into account various classifiers?

RQ2: How does the developed method work against the quality of the classification obtained by different base classifiers for different levels of explained variance?

RQ3: How does the original method of determination of the number of components and features work against the other feature extraction methods known from the literature, taking into account various base classifiers?

RQ4: What are the response times for a query sent to the base classifier for different Industry 4.0 architectures?

### 5.1. Used Classifiers

Below, all classifiers used in the experiments are listed. For each classifier, we provide hyperparameter sets that were considered for fine-tuning.

k-NN—k Nearest Neighbors [53];–number of neighbors: 3, 5, 7,–metrics: Minkowski, Euclidean, Manhattan.SVC—Support Vector Classification/Support Vector Machine [54];–parameter C: 0.1, 1, 10, 100,–kernel: linear, rbf, poly, sigmoid,–gamma: scale, auto.CART—Classification and Regression Trees [55];–criterion: gini, entropy,–splitter: best, random,–maximum depth: 1, 2, 3, ..., 10.GNB—Gaussian Naive Bayes—without parameters [56,57];MLP—Multi-layer perceptron [58];–number of hidden layers: 3, 4, 5, ..., 10,–activation function: identity, logistic, entropy, SOS, Tanh, Linear, Softmax, Exponent,–parameter alpha: 0.00001, 0.0001, 0.001, 0.01, 0.1,–momentum: 0, 0.2, 0.4, 0.6, 0.8, 1.

All hyperparameters were tuned automatically with the statistica software.

For the classification task, the KNN classifier was used with 3, 5, and 7 nearest neighbors. The best results for MLP were obtained with 7 hidden layers, linear activation function, alpha parameter set to 0.01 and momentum of 0.6. For K-NN, the best result was achieved using Euclidean distance and 7 nearest neighbors. For SVC, the best balanced accuracy score was acquired for C = 0.1, rbf kernel and gamma parameter set to auto. For decisions trees CART, the best classification quality was obtained with criterion gini, splitter: best and maximum depth of 7. The best classification results were applied in all experiments.

### 5.2. Setup

The study used a dataset consisting of 40,000 objects. That relates to the assessment of chemical hazards in the pharmaceutical industry. The dataset contains of a total of 63 features most often associated with toxic and physicochemical properties. They include the properties of various active substances that are responsible for the therapeutic effect and auxiliary substances, such as fillers, carriers and solvents, which are used in the production to obtain the appropriate form, hardness or appearance of the drug. Parameters, i.e., concentration, odor intensity, etc., may cause a threat to the health and life of employees. Based on the measured 40,000 different values of features, two experts labeled the data classifying feature vector into one of four risk classes: no risk (33.36 k; 79.5%), low risk (4.84 k; 12.1%), medium risk (1.6 k; 4%), high risk (1.4 k; 3.5%). Due to the unbalanced nature of data, a five-fold stratified cross-validation method was used in three experiments. The Balanced Acurracy Score metric (BAC-*score*) was used to assess the quality of the classification.

### 5.3. Comparison of Classification Accuracy for Various Criteria of Selecting the Number of Components and Features for the PCA Method

The purpose of the experiment 1 is to answer the first research question. For this purpose, a study was carried out to verify whether the developed method for the determination of the number of components and features is at least as accurate as in the case of the K—Kaiser and SP—scree plot criteria. For this reason, the obtained quality results were compared with a case of: (a) no extraction—NO, (b) extraction using only the kaiser criterion K, (c) only scree plot criterion SP and (d) fusion of these two criteria SP+K. The proposed method is marked with ${}^{\prime }$. The results were verified with the Wilcoxon signed-rank test at the statistical significance level *p* = 0.05 comparing the quality obtained by different classifiers between different extraction methods. Results are presented in Table 1 and Figure 4.

The first important conclusion from the experiment is that the quality of the classification when using the proposed PCA’ method is significantly (*p* < 0.05) better than other methods. For each classifier, the proposed criterion allowed obtaining statistically better classification qualities. For each PCA method, four principal components were distinguished. Selected components for the classic PCA model explained 81% of the total variance. For the proposed solution, the percentage of the total variance is approximately 83%. The number of features that remain after the extraction task, depending on the used criterion is: ${\mathrm{PCA}}_{K}$ (n = 25), ${\mathrm{PCA}}_{\mathrm{SP}}$ (n = 24), ${\mathrm{PCA}}_{K+\mathrm{SP}}$ (n = 22) and PCA’ (n = 18). The selected set of features using PCA’ has a better discriminant power than other criteria, which was confirmed by the analysis of the discriminant function. To evaluate the criterion of maximizing, the Wilks lambda coefficient was adopted. After applying the proposed method, in contrast to the classic Kaiser criteria and the scree plot, the features which in the discriminant analysis had a value of Wilks’ partial lambda at the level of statistical significance (0.5 < *p* < 0.7) were rejected. After applying a single criterion, i.e., Kaiser or scree plot, comparable qualities of correct classifications were obtained. Using both criteria in the ${\mathrm{PCA}}_{K+\mathrm{SP}}$ for the SVC, KNN7, and MLP classifiers, the results were better than using only the scree plot. In the task of classifying chemical hazards in drug production, the best quality results were obtained when SVC and MLP methods were used.

### 5.4. Comparison of Classification Accuracy Depending on the Percentage of Explained Variance

The experiment answers the second research question. Since in experiment 1 the used of both K—Kaiser and SP—scree plot criteria for the selection of the number of components gave better results of correct classification than in the case for each of the above-mentioned criteria separately, therefore, the method PCA with criteria K and SP was used in the study. The quality of the classification was tested, adopting the number of components according to the percentage participation in the explained variance. The results were compared with the PCA${}^{\prime }$ method and are shown in Figure 5.

In Figure 5, the horizontal line is the BAC-*score* quality obtained by the given classifier with the extraction carried out by the proposed PCA’ method. It is a constant value as it does not depend on the percentage of explained variance. For the classic PCA, it can be noticed that for each classifier the obtained quality increases and reaches the maximum at 80% limit. This is due to the fact that classification was started with a few features with their entire vector. The developed new criterion for selecting the number of components and features allows to obtain better results than with the classic criteria.

### 5.5. Comparison of Classification Accuracy between Different Feature Extraction Methods

In the last experiment, the results of which are presented in Table 2 and Figure 6, the obtained classification qualities were compared between different methods of feature extraction. On the one hand, the quality of classification was tested based on the original criteria. On the other hand, the proposed approach was used for each method. Such a method was marked with the sign ${}^{\prime }$.

Now, we can conclude that the use of the proposed approach for the determination of the number of components and features in various extraction methods statistically significantly improves the quality of classification for a given example of unbalanced data. These results are interesting as they show that this solution can be used in various extraction methods based on principal components. In any case, after application of the proposed criterion, the quality of all classifiers significantly increases. Among the compared methods, the highest classification qualities were obtained for CCPCA${}^{\prime }$ and GPCA${}^{\prime }$, which are based on component rotation according to class centroids and the use of the stochastic gradient method. The results of the research also confirm that the proposed approach for the determination of the number of components and features in comparison with the K—Kaiser and SP—scree plot criteria for unbalanced data improves the quality of the classification.

### 5.6. Compare Different Real-Time Data Processing Architectures

In this research part, in the AnyLogic simulation environment, the proposed concept was modeled in Figure 3. AnyLogic is a professional environment used by various industrial groups to model very complex architectures. It has many built-in ready libraries and modules, which greatly facilitate the creation of simulations. An important advantage of this software is the ability to combine it with Python scripts. Feature extraction methods and base classifiers have been implemented in Python. The study used the best method of CCPCA extraction and the two best classifiers for the discussed issue: MLP neural network and SVC support vector machine. These methods are selected based on the experiments performed earlier in this article. The logical diagram of the research model is presented in Figure 7.

The database contains the data tested in previous experiments. The research assumed that there are eight pharmaceutical companies. From a sample of 40,000 records, all patterns were randomly divided into eight groups of 5000 records, each to perform a five-fold cross-validation. The learning process was performed on 35,000 records, and testing on 5000, 1000 records for each company. It was assumed that each of the eight companies has one of the eight horizontal management systems, i.e., Big Data, Digital Twins, Cloud Computing, Semantics, ID/Log, IIoT, SOA and Connectivity.

In the developed model, query contains a set of queries to the server about the classification results for the analyzed pattern. The Source function initializes the model. Then, timeStart starts the clock. A block with eight architectures carries out communication processes in each of the enterprises and sends the first records for model testing. This information goes to the Stay block, which is responsible for classification using MLP and SVC. This task is performed on datasets without and after feature extraction. Then, the timeEnd time from sending the query to receiving the classification result is calculated. In the next steps, the process is repeated until all test patterns are used from the set. The hold function does not allow the model to end before the complete testing task is completed.

Figure 8 shows the obtained research results. Decision times are in ms for each system. For the architectures Big Data and Digital Twins, the data processing speed is lower than for other architectures considered. Due to the speed of response, the SVC support vector machine seems to be a better solution than the MLP. Previous experiments showed that the quality of the classification is similar for the above-mentioned classifiers. Feature extraction allows one to carry out the classification process faster and obtain answers faster. Analyzing the obtained results, we can conclude that the response times are stable and fall between 16 ms and 56 ms for MLP and 18 ms and 59 ms for SVC. Therefore, in the best case, to ensure good system performance, we can send from 17 (56 ms) to 62 queries (16 ms) per s.

## 6. Conclusions

The article presents a proprietary method of determining the number of components and features in the extraction task. The method was used in the chemical risk assessment of a dataset with 63 features most often associated with toxic (n = 18), physicochemical properties (n = 28), and network security (n = 17). The number of features that remain after the extraction task, depending on the used criterion, is 18 features most often associated with toxic (n = 6), physicochemical properties (n = 7), and network security (n = 5). Three experiments were carried out to draw the following conclusions:1.A method of estimating the number of components and features was developed for the principal components method. The developed method can be used to extract feature spaces between which there are also nonlinear relationships. An important conclusion is that it was possible to extract from 63 key features for the chemical risk classification task, most often associated with toxic (n = 6), physicochemical properties (n = 7), and network security (n = 5). Key features related to cybersecurity are: (a) Number of Packets Lost, (b) Incorrect Logins, (c) Incorrect Sensor Responses, (d) Increased Email Spam, (e) Excessive Traffic in the Computer Network. The extracted features explain 84% of the total variance for the CCPCA method and 81% for PCA. For the Kaiser criterion and the factorial scree, the total variance is 78%. The cybersecurity component explains 8% of the total variance. It can be concluded that cybersecurity features help increase the quality of the classification. This is practical because the risk of a threat increases with some redundant network traffic. Quick response in this regard can prevent the threat of a cyber-attack by paying attention to the factors identified.2.Experiment 1 showed that the estimation of the dimensionality of the features for PCA based on the correlation method and discriminant power allows obtaining a better quality of classification by about 4% compared to no extraction and by about 2.3% compared to using only the scree factor and about 2% better than the Kaiser criterion. The best classification qualities were obtained for the MLP neural network.3.In experiment 2, it was shown that for different percentages of explained variance, that is, for different feature spaces after extraction, the developed method allows to obtain better qualities of correct classifications.4.In experiment 3, tests were carried out for other extraction methods based on the main sample method, i.e., CCPA, GPCA, and KPCA. Interesting results were obtained in this experiment. It turned out that after applying the proprietary method for each of the methods, the quality of the classification improved. The best results were obtained for the CCPCA, and GPCA extraction methods. In the case of CCPCA, it is conditioned by the fact that the rotation of features is carried out according to class centroids, which gives better discrimination of feature spaces. In the case of GPCA, the angle of rotation is adjusted as a result of optimization using the stochastic gradients method. It also allows better matching of features to a given component.5.In experiment 4, the AnyLogic simulation environment was proposed to virtualize various systems used in Industry 4.0. By integrating this environment with Python, the quality of the classification was assessed by analyzing the classifier response times depending on the systems used. Using the built-in ready-made data processing modules in AnyLogic, the CPS systems response time was assessed depending on the method of classification and feature extraction. It has been shown that companies using Big Data and Digital Twins obtain faster feedback from the base classifiers. Big Data gives the best data processing speeds due to the management of the scale, variety, and speed of data. This technology allows streaming data in real-time. Digital Twins turned out to be the second-fastest of data processing. This result can be obtained by using a digital replica of physical objects, processes, and systems. Thanks to the virtualization of reality through digital mapping of a physical object, we can process data faster in real-time. It also allows you to constantly update the state of objects and processes. Thanks to this approach, it is possible to increase the speed of sending an inquiry and receiving a decision on the degree of chemical risk. SOA and Connectivity systems deserve attention. The reason for the slower SOA response is its architecture. It is similar to distributed objects but at a higher level of abstraction. The services themselves are often implemented based on different technologies and made available using an independent communication protocol. Systems of this type optimize the processing speed worse than in the case of Big Data or Digital Twins. We can also notice that using the local 5G campus network in Connectivity technology is also not a better solution in terms of efficiency than Big Data or Digital Twins.

The results obtained in the research are promising. In the course of research and analysis of the literature, it is possible to outline further research directions in the above-mentioned range:1.Increasing the number of cybersecurity-related features and simulating attacks, such as Malware, Man in the Middle, Cross-site scripting, Phishing, DDoS, SQL Injection, Ransomware, Malvertising. Such a solution will allow to the estimation of the characteristics sensitive to such attacks, which will increase the quality of the classification and the discriminatory power of the risk assessment of chemical hazards.2.Carrying out multi-criteria optimization evaluating various criteria influences the quality of the classification. These criteria are the acquisition cost of the traits and the Balance Accuracy.

## Figures and Tables

**Figure 1 sensors-21-05753-f001:**
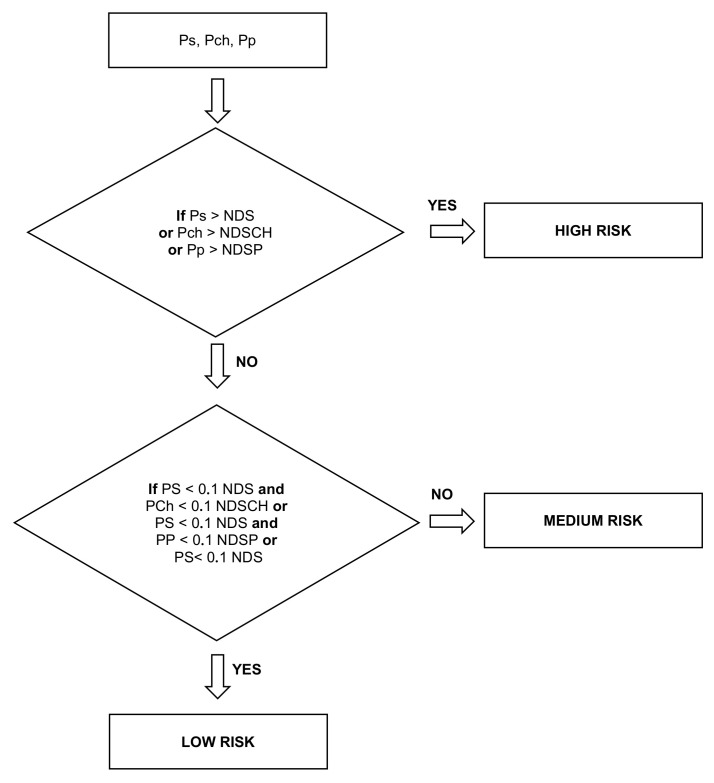
A three-step algorithm for estimating the risk associated with exposure to harmful chemical agents in the air during the production of drugs.

**Figure 2 sensors-21-05753-f002:**
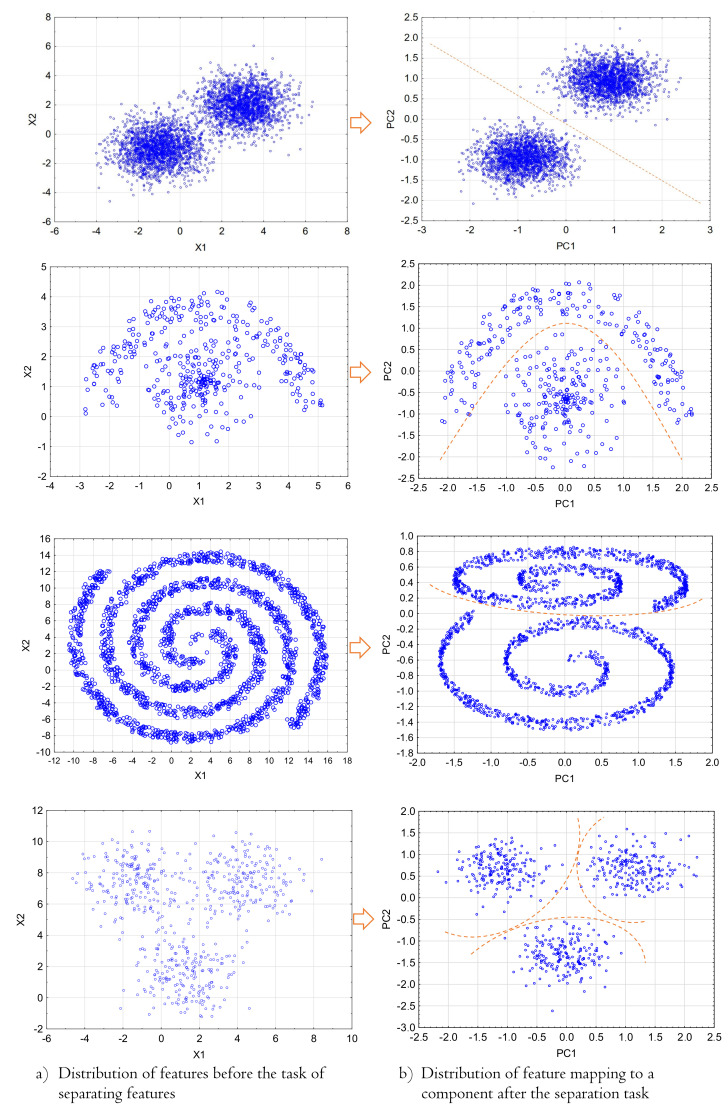
Chart showing the determining method of the number of features and components in the case of linear and nonlinear separation of features.

**Figure 3 sensors-21-05753-f003:**
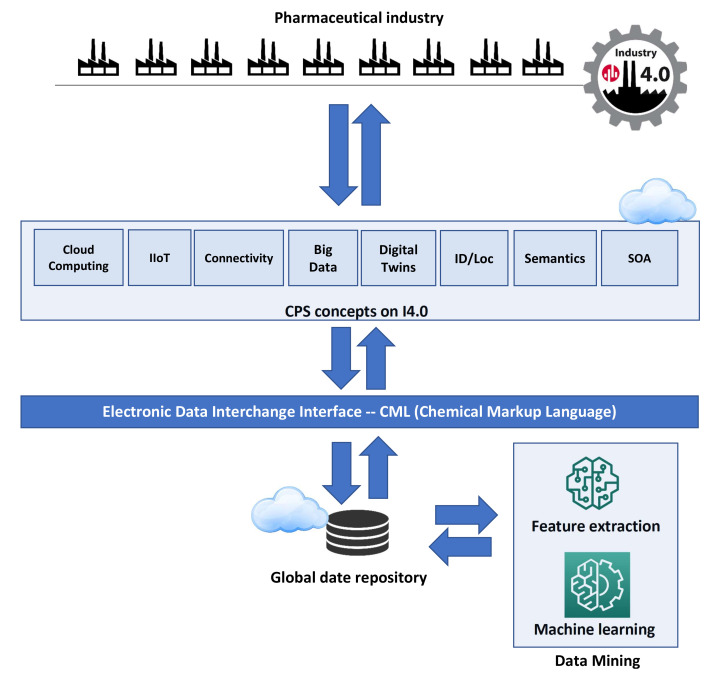
Chemical risk classification model in the pharmaceutical industry.

**Figure 4 sensors-21-05753-f004:**
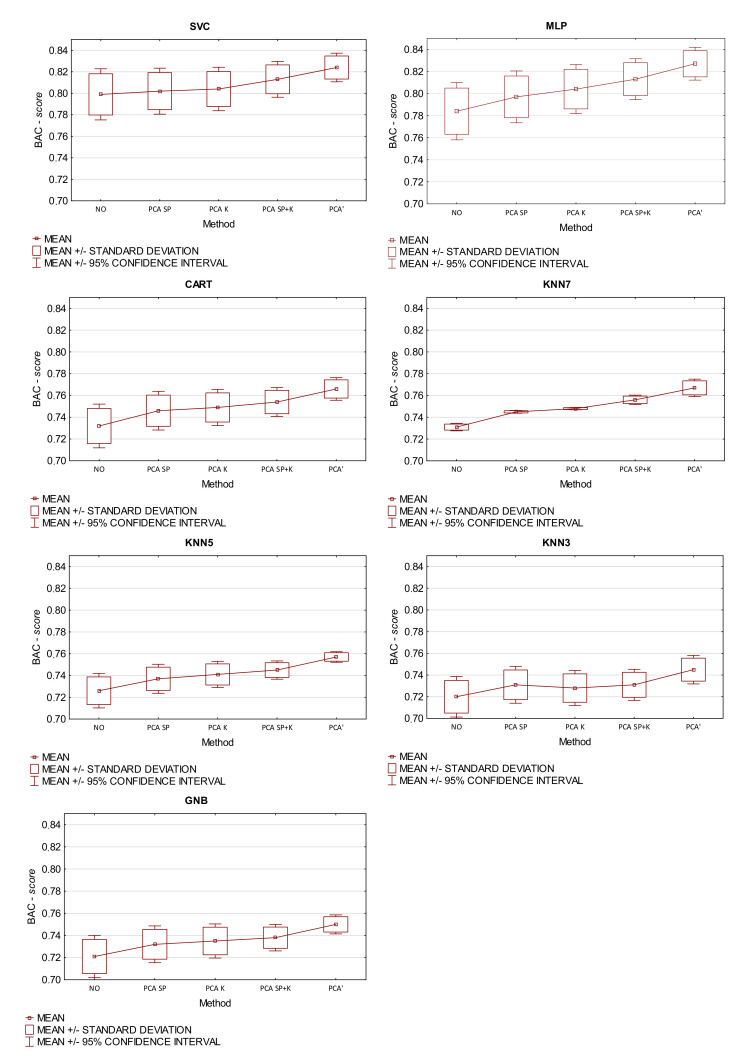
Statistics of the experiment 1 when the proposed method is used to estimate the number of components and features with the BAC-*score* metric. The lines contain extraction methods and the columns contain classifiers.

**Figure 5 sensors-21-05753-f005:**
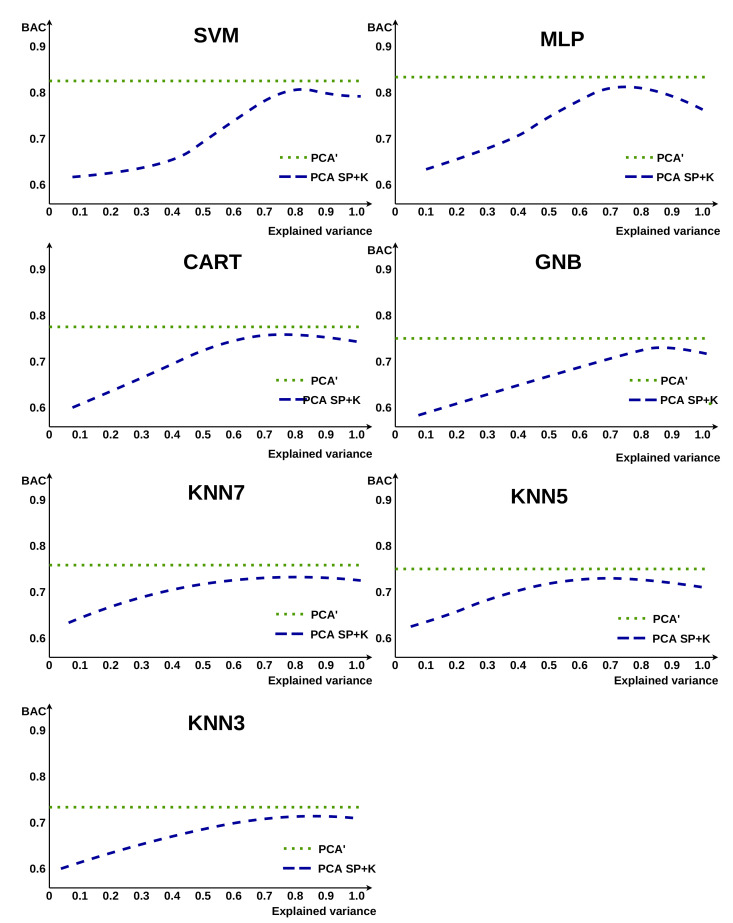
The results of the experiment 2, classification quality depending on the percentage of explained variance for PCA with the BAC-*score* metric. The results were compared with the PCA${}^{\prime }$ method.

**Figure 6 sensors-21-05753-f006:**
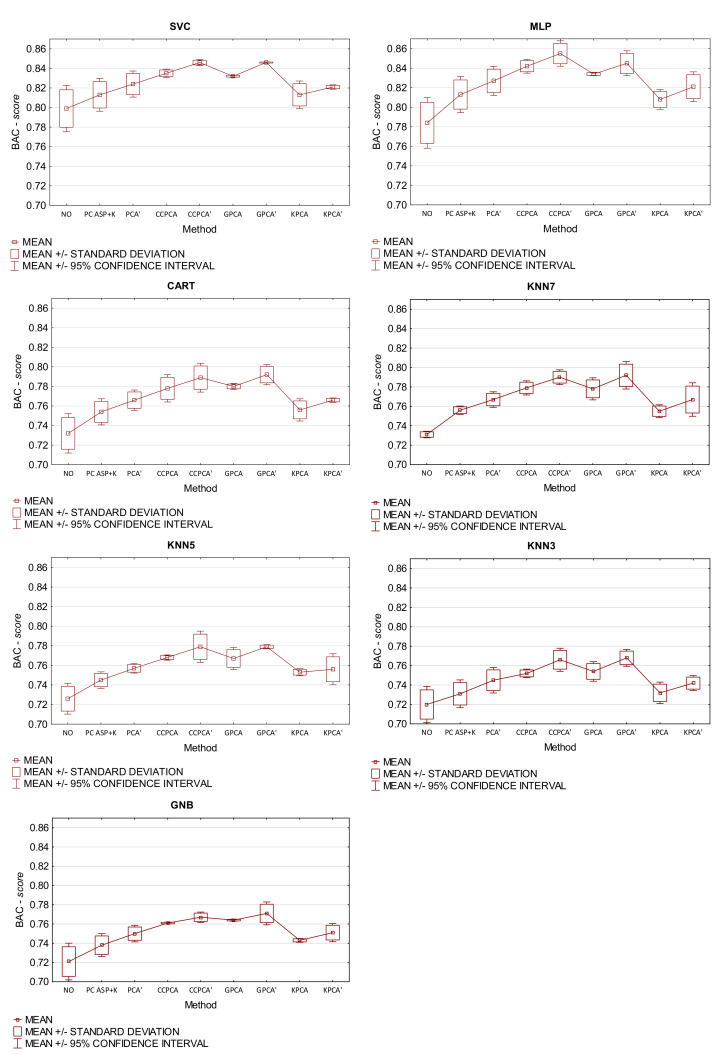
Statistics of experiment 3 for the case of using the proposed method of determining the number of components and features in various extraction methods with the BAC-*score* metric.

**Figure 7 sensors-21-05753-f007:**
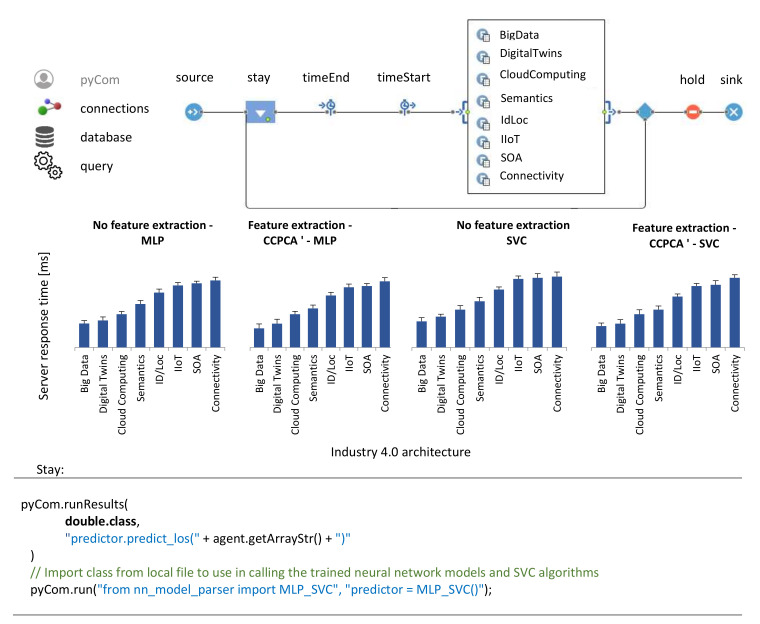
The experimental model developed at AnyLogic.

**Figure 8 sensors-21-05753-f008:**
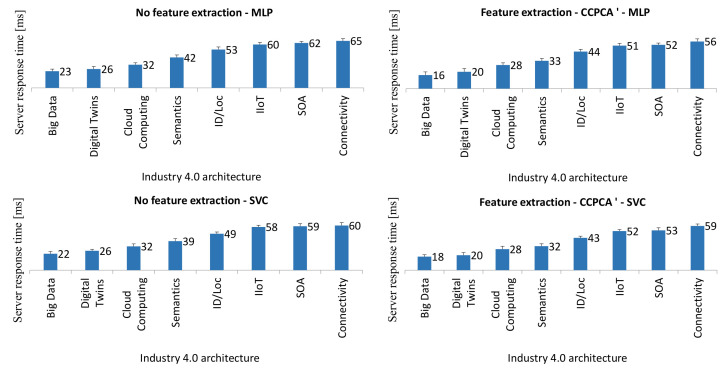
Graph of the dependence of times of obtained responses depending on the Industry 4.0 architecture.

**Table 1 sensors-21-05753-t001:** Results of the experiment 1 when the proposed method was used to estimate the number of components and features with the BAC-*score* metric. The lines contain extraction methods and the columns contain classifiers.

metods	SVC	KNN7	KNN5	KNN3	MLP	CART	GNB
1. NO	0.799	0.731	0.726	0.720	0.784	0.732	0.721
	${}^{-}$	${}^{-}$	${}^{-}$	${}^{-}$	${}^{-}$	${}^{-}$	${}^{-}$
${2.\mathrm{PCA}}_{\mathrm{SP}}$	0.802	0.745	0.737	0.731	0.797	0.746	0.732
	${}^{1}$	${}^{1}$	${}^{1}$	${}^{1}$	${}^{1}$	${}^{1}$	${}^{1}$
${3.\mathrm{PCA}}_{K}$	0.804	0.748	0.741	0.728	0.804	0.749	0.735
	${}^{1}$	${}^{1}$	${}^{1}$	${}^{-}$	${}^{1}$	${}^{1}$	${}^{1}$
${4.\mathrm{PCA}}_{\mathrm{SP}+K}$	0.813	0.756	0.745	0.731	0.813	0.754	0.738
	${}^{1,2}$	${}^{1,2}$	${}^{1}$	${}^{1}$	${}^{1,2}$	${}^{1}$	${}^{1}$
5. PCA’	0.824	0.767	0.757	0.745	0.827	0.766	0.750
	${}^{1-4}$	${}^{1-4}$	${}^{1-4}$	${}^{1-4}$	${}^{1-4}$	${}^{1-4}$	${}^{1-4}$

**Table 2 sensors-21-05753-t002:** Results of experiment 3 for the case of using the proposed method of determining the number of components and features in various extraction methods with the BAC-*score* metric. The lines contain extraction methods and the columns contain classifiers.

metods	SVC	KNN7	KNN5	KNN3	MLP	CART	GNB
1. NO	0.799	0.731	0.726	0.720	0.784	0.732	0.721
	${}^{-}$	${}^{-}$	${}^{-}$	${}^{-}$	${}^{-}$	${}^{-}$	${}^{-}$
2. ${\mathrm{PCA}}_{K+\mathrm{SP}}$	0.813	0.756	0.745	0.731	0.813	0.754	0.738
	${}^{1}$	${}^{1}$	${}^{1}$	${}^{-}$	${}^{1}$	${}^{1}$	${}^{1}$
3. PCA${}^{\prime }$	0.824	0.767	0.757	0.745	0.833	0.766	0.750
	${}^{1,2,8}$	${}^{1,2,8}$	${}^{1,2}$	${}^{1,2,8}$	${}^{1,2,8,9}$	${}^{1,2}$	${}^{1,2}$
4. CCPCA	0.835	0.779	0.768	0.752	0.842	0.778	0.761
	${}^{1-3,8,9}$	${}^{1-3,8,9}$	${}^{1-3,8,9}$	${}^{1,2,8}$	${}^{1,2,8,9}$	${}^{1-3,8,9}$	${}^{1-3,8}$
5. CCPCA${}^{\prime }$	0.846	0.790	0.779	0.766	0.855	0.789	0.767
	${}^{1-4,6,8,9}$	${}^{1-4,6,8,9}$	${}^{1-4,6,8,9}$	${}^{1-4,6,8,9}$	${}^{1-4,6,8,9}$	${}^{1-4,8,9}$	${}^{1-3,8,9}$
6. GPCA	0.832	0.778	0.767	0.754	0.834	0.780	0.764
	${}^{1,2,8,9}$	${}^{1-3,8,9}$	${}^{1,2,8,9}$	${}^{1,2,8,9}$	${}^{1,2,8,9}$	${}^{1-3,8,9}$	${}^{1-3,8,9}$
7. GPCA${}^{\prime }$	0.846	0.792	0.779	0.768	0.845	0.792	0.771
	${}^{1-4,6,8,9}$	${}^{1-4,6,8,9}$	${}^{1-4,6,8,9}$	${}^{1-4,6,8,9}$	${}^{1-3,6,8,9}$	${}^{1-4,6,8m9}$	${}^{1-3,6,8,9}$
8. KPCA	0.813	0.755	0.753	0.732	0.808	0.756	0.743
	${}^{1}$	${}^{1}$	${}^{1}$	${}^{1}$	${}^{1}$	${}^{1}$	${}^{1}$
9. KPCA${}^{\prime }$	0.821	0.767	0.756	0.742	0.821	0.766	0.751
	${}^{1}$	${}^{1,2,8}$	${}^{1,2}$	${}^{1,2}$	${}^{1,8}$	${}^{1,2}$	${}^{1,2}$

## Data Availability

Not applicable.

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
