# Peer review of "Application of Feature Extraction Methods for Chemical Risk Classification in the Pharmaceutical Industry"

_sensors, 2021, doi:10.3390/s21175753_

Round 1

Reviewer 1 Report

COMMENTS:

  1. It is worth shortening somewhat the descriptive part of Industry 4.0, which contains many well-known statements that do not add anything to the article's essence (lines 95-129, 222-228, 233-236).
  2. Usually, the "Introduction" or "Related work" sections end with a brief description of the proposed approach, indicating its novelty. However, the end of Section 3 of this paper shows the research goals without identifying distinctive features from other scientific studies in the field.
  3. The description of the methodology lacks a visual description in the form of pseudocode or flowchart.
  4. It is unclear why Tables 1-2 and Figures 5-7 show the results for kNN with different values of the hyper-parameters, and for the other methods the hyper-parameters values are not even indicated.
  5. Probably better to replace the word "proprietary" with "proposed" (lines 257, 311, 365, etc.).
  6. In Figure 9, there is no clear dependence of the time on the number of the query sent; therefore, these results can be presented in a more visual form, for example, in the form of a histogram (Vertical axis is "Server response time", horizontal - Industry 4.0 architecture).
  7. In conclusion, the statement "The developed method fills the existing gap in the estimation of the number of components and features" (lines 501-502) is too categorical.

Author Response

Dear Editor,

Please find enclosed a revised version of the manuscript (sensors-1235505). Thank you very much for important comments on our work.

We took into account all comments from reviewers. If the article meets the expectations of the reviewers, we kindly ask you to publish it.  We are grateful to the effort done by the reviewers and Editor to provide us with valuable feedback. We have found the suggestions and comments useful for preparing the second version of this article.

Best regards,

Mariusz Topolski

Reviewer 2 Report

This paper presents a research about the analysis of chemical hazards arising in the pharmaceutical industry during drug production. In particular, the Author proposes a novel method for determining the number of components and features of them. From the point of view of the Reviewer, the scope of this paper is more oriented towards the management of data than to the development of a new sensor. Therefore, may be this publication must be transfer to another Journal dealing with scientific aspects of data management. In addition, the Reviewer would like to recommend the following revisions to the Author.

  1. Please consider in changing the title of the paper to “Application of Feature Extraction Methods for Chemical Risk Classification in the Pharmaceutical Industry”.
  2. The Abstract must be re-written. Almost half of it is discussing about general facts, please try to be more direct. Remember that the Abstract is one of the most important sections of the paper. Basically, if the Abstract section is not clear, the reader won’t continue reading the paper. Abstract must be a stand-alone description of the paper. In this way, Abstract Section must be constructed based on a motivation, approach, results, and conclusions written in around 200 or 300 words (depending on the Journal). Please revise the Abstract Section.
  3. The Introduction Section is very confusing, it seems that the Author is describing very broadly the problem that wants to address. Introduction section must be improved. For example, by the end of the Introduction Section, the main contribution of this paper to the Sensors Journal must be clearly stated based on the objectives of the research. Please revise the Introduction Section.
  4. Figures 2, 3, and 4 must be included in the paper right next where they are mentioned. Please revise this.
  5. There are a lot of acronyms in the manuscript. Please consider including a list of acronyms.
  6. Conclusions are long and confusing. Please try to reformulate them using bullet points. Please be more direct when documenting conclusions.
  7. Please include DOI to every reference.

Author Response

(The authors gave the same response as above.)

Round 2

Reviewer 2 Report

It seems that the Author has addressed every recommendation made by the Reviewer. The paper is now suitable for publication.